# Hydrangea DNA Methylation Caused by pH Substrate Changes to Modify Sepal Colour is Detected by MSAP and ISSR Markers

**Julio Y. Anaya-Covarrubias [1], Nerea Larranaga [1], Norma Almaráz-Abarca [2], Martha Escoto-Delgadillo [1] , Ramón Rodríguez-Macías [1] and Martha I. Torres-Morán [1,\*]**

[1]   Centro Universitario de Ciencias Biológicas y Agropecuarias, Universidad de Guadalajara, km 15.5 carretera a Nogales, 45101 Zapopan, Mexico; persefone@hotmail.com (J.Y.A.-C.); nerelarra_@hotmail.com (N.L.); martha.escoto@academicos.udg.mx (M.E.-D.); ramonrod@cucba.udg.mx (R.R.-M.)

[2]   Centro Interdisciplinario de Investigación para el Desarrollo Integral Regional, Instituto Politécnico Nacional, 34220 Durango, Mexico; noralab@yahoo.com

\*   Correspondence: isabel.torres@academicos.udg.mx; Tel.: +33-3440-1252

**Abstract:** The hydrangea (*Hydrangea macrophylla* (Thunb). Ser.) is an ornamental species with great market potential. It is known for its ability to change the colour of its inflorescence, according to the pH of the culture substrate. The molecular mechanisms that underlie these changes are still unclear. It is known that epigenetic mechanisms, such as DNA methylation, play an important role in genetic expression, so they could be responsible for this phenomenon in hydrangea. In the present study, the molecular markers ISSR (Inter Simple Sequence Repeat) and MSAP (Methyl-Sensitive Amplification Polymorphism) were used to detect molecular changes in the genome of hydrangea plants that were cultivated under different pH levels to modify the colour of the sepals. The results showed a correspondence between the methylation signal measured with MSAP and amplification ISSR patterns when compared before and after the modification of pH culture substrates. These results suggest that DNA methylation might be involved as a molecular mechanism underlying the colour change of hydrangea sepals in response to a differential pH in the substrate. In addition, the results pave the way to study the relationship between DNA methylation and ISSR marker profiles.

**Keywords:** Hydrangea; colour of sepals; pH; DNA methylation; MSAP; ISSR

## 1. Introduction

The genera *Hydrangea*, native to Japan, from the Hydrangeae family, was divided in two sections: *Hydrangea* McClint. and *Cornidia* Engl., which englobe 25 species in total and numerous subspecies [1]. In addition, many cultivars have been developed and named as species, which complicates the taxonomy of this family [2]. Many of these species are marketed as flower of court, due to its long lasting characteristic umbelliform cyme inflorescences [3]. The inflorescences are complex and polymorphic, since they are composed of small fertile flowers and other sterile with large coloured sepals, which are often much more visible than the petals [4–6]. Most species of the genus are bushes, but some arborescent forms, such as *H. japonica,* or evergreen climbers, such as *H. petiolaris,* are included. According to the University of Tenesse [7], hydrangea was the second most popular deciduous shrub across all horticultural markets in 2014, with more than 10 million plants sold ($91.2 million) in the USA. Many hydrangea cultivars have descended from *Hydrangea macrophylla* (Thunb.) Ser. [8]. These species' ability to change the colour of the sepals as a consequence of the pH in the culture substrate is

largely known in the flower field, although the molecular mechanisms that underlie these changes are still not clear [9].

Most of the flowering plants accumulate pigments in the vacuole, which are predominantly secondary metabolites of the flavonoids pathway. The most widely distributed floral pigment are the anthocyanins, which are big polar molecules that are transported to the vacuole for proper biological functioning [10–12]. Biosynthetic and regulatory genes mediate the biosynthesis of the almost 20 different types of anthocyanin [10,13–15]. The variety of colours from blue to purple and red in hydrangea sepals are just due to the anthocyanin delphinidin 3-glucoside in combination with other co-pigments quinic acid esters, such as: chlorogenic acid, neochlorogenic acid, and 5-O-p-coumaroyl quinic acid, and ion $Al^{3+}$. The composition analysis of protoplast extracts that were derived from blue and red sepals showed that the molar ratios of neochlorogenic acid, 5-O-p-coumaroylquinic, and $Al^{3+}$ to delphinidin 3-Oglucoside were much higher in the blue cells than in the red cells [12,16–20]. Aluminum (Al) is soluble in acid soil (below pH 5.0) as a toxic form, Al3+, considered to be a limiting factor for many crops production. The primary symptom is the inhibition of root growth. The plants have developed different strategies for developing Al tolerance, like the secretion of organic anions from root tips or Al protein transporters [21]. Different Al transporter have already been identified and characterized in hydrangea: HmVALT and HmPALT1, both member of the aquaporin family (only the last only expressed in the sepals) or HmPALT2, an anion permease, whose transcript was expressed in all tissues of hydrangea plants [22,23]. In addition, Chen et al. [24] revealed that 4287 genes in the roots and 730 genes in the leaves of hydrangea were up-regulated, while Al exposure down-regulated 236 genes in the roots and 719 genes in the leaves, after analyzing the transcriptome. In addition, the relation between DNA methylation and colour expression has been described in other species. In *Zea mays*, the methylation or demethylation of P1-Bh y P1-wr genes modifies the accumulation of anthocyanin pigments in corn grain [25,26]. In *Malus domestica*, hypermethylation in the MdMYB10 promoter causes a striped pigmentation in the fruit due to an increase in the concentration of anthocyanins [27]. Alternatively, in the ornamental *Nelumbo nucifera*, different levels of methylation on the promoter sequence of the anthocyanindin synthase gene seem to result in the different red or white flower colouration [28].

Methyl-Sensitive Amplification Polymorphism (MSAP) is a tool that is based on the Amplified Fragment Length Polymorphism (AFLP) technique [29]. The methodology requires the use of restriction enzymes that detect methylated sites. Quantitative differences in amplification indicate variation in the global DNA methylation pattern of individuals. MSAPs help to identify the differences in methylated genomic regions within and between populations with different genetic backgrounds, as well as in plants that were grown under different environmental conditions [30,31]. The advantages of this technique are that it is not necessary to previously know the nucleotide sequence, it has a high level of reliability, and it is easily reproducible and highly polymorphic [32]. On the other hand, Inter Simple Sequence Repeat (ISSR) are molecular markers, based on microsatellite regions, which have been used in plants to register genetic fingerprints, phylogenetic analysis, diversity and genetic variability, and the identification of cultivars [33–35]. They have also been used to document genetic fidelity in plants that come from micropropagation [36,37]. It is a simple and fast method that combines most of the advantages of microsatellites (SSR) and amplified fragment length polymorphism (AFLP) to the universality of random amplified polymorphic DNA (RAPD) [33,35].

In this work, the colour change of the sepal of 28 hydrangea plants was evaluated after being treated with different culture conditions by modifying the pH. The genomic DNA that were extracted from the plants before and after the treatment were analyzed with MSAP to report the possible relationship with methylation pattern changes and with ISSR to verify the genome stability after the substratum change. The documentation of the molecular mechanism that allows for sepal colour change as a consequence of pH substratum can expand the knowledge in the management of hydrangeas plants.

## 2. Materials and Methods

### 2.1. Plant Material

Twenty-eight plants of *Hydrangea macrophylla* (Thunb.) Ser. were obtained from terminal cuttings purchased in the commercial house Schreurs from its production site in Ecuador. Cuttings have six leaves each and 15 cm long of stem. Plants have a rooting period of 10 days under greenhouse, after which the plants were divided in two treatments of 14 plants. Afterwards, they were placed in 20.32 cm pots with two kinds of substratum; a substratum that was composed by 50% organic soil and 50% fine pumice (AL) producing Alkaline pH, and a substratum composed by 60% peat moss and 40% fine pumice (AC) producing Acid pH. The pruning of these plants was carried out to favor the transition of the phenological stages from bud to flowering

### 2.2. Treatments

The comparison was carried out in two cultivation cycles consisted in make change the pH of substratum AL and AC. During the first cycle the 28 plants were treated under the same conditions. Once the first cycle was over, a second cycle begins with deferral pH conditions. Briefly, the plants were established for 30 days, separated into two groups of 14 plants each: AL (alkaline group) and AC (acid group). They were both analyzed (colour, pH and DNA) after flowering. For the AL treatment, an application of sodium hydroxide (NaOH) concentrated at 1% applied as a drench. For the AC treatment, an application of phosphoric acid ($H_3PO_4$) was used at concentration of 2% and applied in drench. All of the plants were fertilized monthly with urea in solution 0.04% + plus fulvic acids (Leonardita) at 0.025% 5 mL per pot, and Bayer® foliar fertilizer Bayfolan forte was used as microelements at 0.03% concentration as foliar application.

### 2.3. pH and Colour Registration

pH was registered in all plants substratum once during both cycles while using the dilution method 1:2 reported by Torres et al. [38]. pH averaged values were compared with a t-test for the first and second cycle and for the AL and AC group. The colour of the sepals was measured also once during both cycles while using a Chroma-meter CR-400 colourimeter and the CIE system, which relies in the measurement of the vectors X = hue, Y = clarity or luminosity, and Z = saturation. These parameters were used to calculate u' and v' following these equations: u´= 4X/(X + 15Y + 3Z) and v´= 9Y/(X + 15Y + 3Z) to plot them in a two dimension graph. All of the plants were also photographed.

### 2.4. DNA Extraction

After the development of the inflorescence in each cycle, genomic DNA was extracted from leaf tissue with the Nucleon PhytoPure kit from GE Healtcare Life Sciences® The extraction was performed according to the supplier's instructions.

### 2.5. Methylation Sensitive Amplification Polymorphism (MSAP)

For MSAP, the methodology was performed, as reported by Baurens et al. [39]. For each individual, two reactions were carried out, one for the enzyme HpaII and one for MspI. Briefly, a restriction test with gDNA (100 ng), 1X buffer MULTI-CORE, BSA 2 µg, 0.5 µL EcoRI (5 U/µL), and 0.5 µL of MspI or HpaII (5 U/µL) was performed in a reaction of 20 µL. The digested DNA H-Linker and E-Linker specific oligonucleotides (Table 1) were adapted to a final concentration of 0.5 µM of each of the adapters, T4 ligase (4 U), 1X buffer T4 ligase in a final reaction of 35 µL. Subsequently, preselective amplification was performed through PCR of the ligation product. The PCR reaction was carried out in a Techne® flexigene thermal cycler under the following conditions: 2 µL DNA of the ligation product was taken, 4 µL 1X Buffer Taq DNA polimerasa, dNTPs (0.3 mM), TaqPol (2.5 U), first ECO + A (0.5 µM), first HPA + A (0.5 µM), and $MgCl_2$ (2.5 mM) in a final reaction of 20 µL. The temperature profile was: an initial

step of 5 min. at 94 °C, 20 cycles of 1 min. at 94 °C, 1 min. at 56 °C and 2 min. at 72 °C, and a final step of 5 min. at 72 °C Finally, the product of the preselective amplification was made a 1:20 dilution and a selective amplification was carried out by means of PCR. The conditions are described below: first ECO + AC (0.5 μM), first HPA2ATG (0.5 μM), dNTPs (0.3 mM), 1X Buffer Taq DNA polymerase, MgCl2 (2.5 mM), and TaqPol 0.4 μL (2.5 U/μL). In a final reaction of 20 μL. The temperature profile was: an initial step of 2 min. at 94 ° C, 40 cycles of 30 s at 94 °C, 30 s at 65 °C and 1 min. at 72 °C plus 30 cycles of 30 s at 94 °C, 30 s at 56 °C, and 1 min. at 72 °C, followed by a final step of 2 min. at 72 °C. The enzymes, buffer, and BSA used in the restriction, binding, and amplification tests belong to Promega®. Table 1 describes the primers used and adapters. PCR solutions were visualized on 6% SDS-polyacrylamide gels stained with silver salts following Sanguinetti et al. [40].

**Table 1.** Name, nucleotide sequence and function of each primer and adapters used in the Methyl-Sensitive Amplification Polymorphism (MSAP) and Inter Simple Sequence Repeat (ISSR) analyses.

| Name | Sequence | Technique |
|---|---|---|
| ELINK1 | CTCGTAGACTGCGTACC | MSAP adapter |
| ELINK2 | AATTGGTACGCAGTCTAC | MSAP adapter |
| HLINK1 | GATCATGAGTCCTGCT | MSAP adapter |
| HLINK2 | CGAGCAGGACTCATGA | MSAP adapter |
| HPA + A | ATCATGAGTCCTGCTCGGA | MSAP primer pre-selective PCR |
| ECO + A | GACTGCGTACCAATTCA | MSAP primer pre-selective PCR |
| ECO + AC | GACTGCGTACCAATTCAC | MSAP primer selective PCR |
| HPA2ATG | ATCATGAGTCCTGCTCGGATG | MSAP primer selective PCR |
| MAO | CTCCTCCTCCTCRC | ISSR primer |
| OMAR | GAGGAGGAGGAGRC | ISSR primer |
| AW3 | GTGTGTGTGTGTRG | ISSR primer |
| 898 | CACACACACACARY | ISSR primer |
| BECKY | CACACACACACACAYC | ISSR primer |
| 844 | CTCTCTCTCTCTCTRC | ISSR primer |

### 2.6. Inter-Simple Sequence Repeat

For the ISSR amplification each PCR reaction, which was carried out in a Techne® flexigene thermocycler, contained PCR buffer 1X, 2.5 mM $MgCl_2$, 0.8 μM of primer, 0.25 mM each dNTP, 2.5 U of Taq. DNA polymerase (Promega®), and 2.5 ng of genomic DNA in a final volume of 20 μL. Table 1 indicated the sequence of the ISSR primer used. The temperature profile was: an initial step of 3 min. at 95 °C, 40 cycles of 45 s at 95 °C, 45 s at 52 °C and 1.5 min. at 72 °C, and a final step of 10 min. at 72 °C. The PCR solutions were visualized on 1.25% high resolution agarose gels (Promega®) that were stained with ethidium bromide 0.5 μg/mL.

### 2.7. Statistical Analysis

MSAP resulting bands were analyzed while using the r package MSAP [41]. In addition, the number of methylated sites was counted, based on the band patterns and the recognition sites of each MSAP enzymes, to calculate the percentage number of methylation sites for each group, as well as an AMOVA (Analyses of Molecular Variance). Averaged values were compared by a t-test for the first and second cycle and for the AL and AC group. In addition, $X^2$ and variance analysis ANOVA was performed to relate these numbers before and after the pH substrate change. ISSR bands were counted and then collected in two binary presence/absence matrices (1 and 2 cycle) that were used to produce similarity matrices while using the Jaccard index [42]. Based on them an UPGMA tree (Unweighted Pair Group with Arithmetic Average) for each cycle was constructed, as well as a comparison between ISSR and MSAP data while using the Mantel test, all analyses included in NTSYSpc® ver. 2.21 software.

## 3. Results

### 3.1. pH and Colour Registration

Figure 1 indicates pH registrations for all photographed plant before and after the pH substratum change (first and second cycle for AL and AC groups). Figure 2 shows mean and standard deviation for pH. Differences among AL and AC during the first cycle were significant with *p* value < 0.05, but not *p* value < 0.01. pH values at the beginning were already acid (around 4.8). Differences among groups during the second cycle were very clear, while the AL reached values around 9.3, the AC dropped to values close to 2.5. First cycle hydrangeas, which were cultivated under the same conditions, produced different bloom phenotypes in the form, size, and colour of sepals. Sepal colour varied from white and blue, to the magenta and purple (Figure 1). Blooms that were produced during the second cycle showed hue changes in all plants and changes in the pigmentation pattern of the white phenotypes. Nevertheless, u' and v' during the first and second cycle for AL and AC groups graph in Figure 3, did not show a difference pattern before and after the treatment.

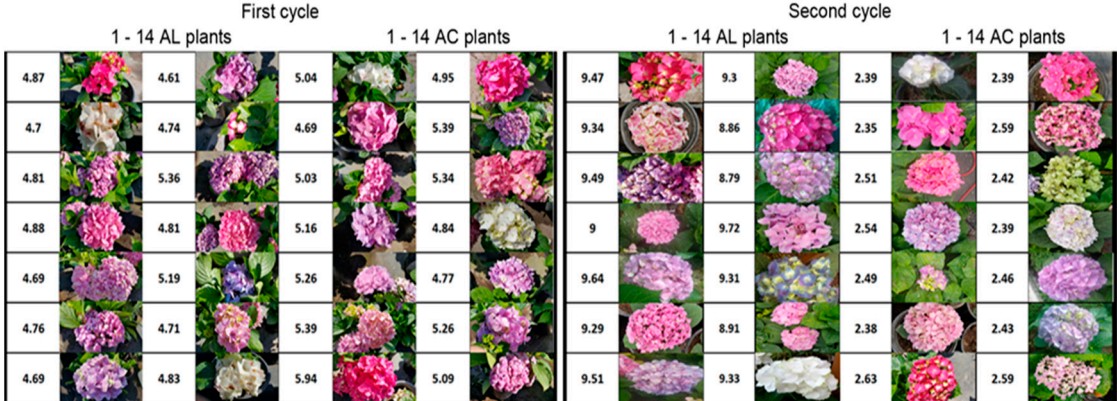

**Figure 1.** Substratum pH registrations and photographed bloom produced during the first and second cycle for the alkaline (AL) and acid (AC) groups.

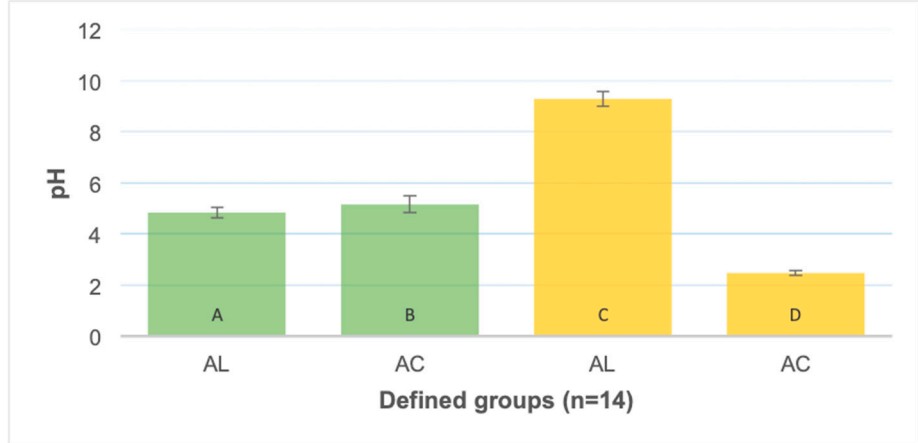

**Figure 2.** Mean and standard deviation pH values during the first (green) and second (orange) cycle for the AL and AC groups. Letters indicate significant differences after pair t-test comparisons among groups within and between cycles for a *p*-value of 0.05.

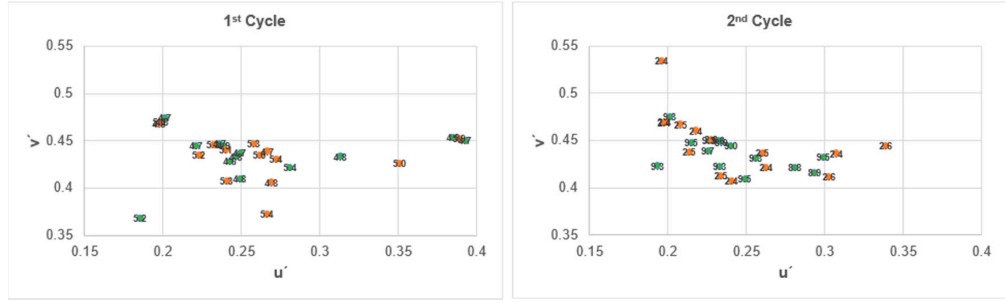

**Figure 3.** Colour u′ and v′ values obtained for each plant during the first and second cycle for the AL (green) and AC (orange) groups. Data labels correspond to each plant pH recorded values.

### 3.2. MSAP

The results obtained with the MSAP package showed the proportion of Methylation susceptible loci (MSL) that are used to assess epigenetic variation and Nonmethylated loci (NML) used to assess genetic variation [41]. Table 2 shows the values.

**Table 2.** Proportion of methylation susceptible loci for each treatment and cycle.

| Methylation Levels | Proportion | | | |
| --- | --- | --- | --- | --- |
| | AL 1st Cycle | AC 1st Cycle | AL 2nd Cycle | AC 2nd Cycle |
| HpaII+/MspI+ (Unmethylated) | 0.47727 | 0.5292 | 0.3312 | 0.2305 |
| HpaII+/MspI− (Hemimethylated *) | 0.08442 | 0.1136 | 0.1818 | 0.2565 |
| HpaII−/MspI+ (Internal cytosine methylation *) | 0.24351 | 0.1753 | 0.3182 | 0.3539 |
| HpaII−/MspI− (Full methylation or absence of target **) | 0.19481 | 0.1818 | 0.1688 | 0.1591 |

* MSL = Hemimethylated + Internal cytosine methylation. ** NML= Full methylation or absence of target.

Significant differences were found in the AMOVA performed for MSL and NML significative at $p < 0.0001$ and $p = 0.001$, respectively. An increase in the proportion of hemimethylated and internal cytosine methylation sites in the second cycle reveals different epigenetic status for plants that are subjected to a change in pH. Figure 4 indicates the mean and standard deviation of number of methylated sites (first and second cycle for AL and AC group) for each group. For both alkaline and acid treatments, the number of methylated sites increased. For the AL group, the total methylated sites varied from 99 to 154, while in the AC the variation was from 88 to 204. The ANOVA and $X^2$ test analysis between the first and second cycle also indicated significant differences.

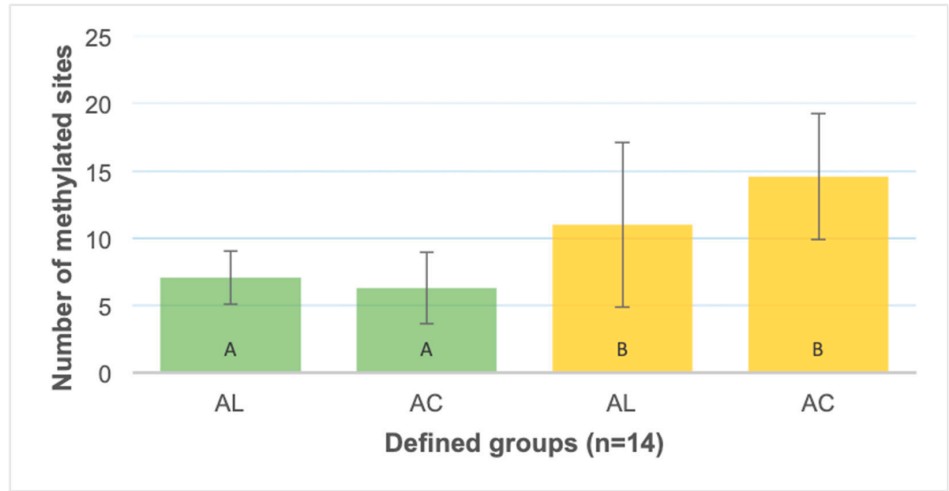

**Figure 4.** Mean and standard deviation of methylated sites encountered after MSAP analysis of DNA during the first (green) and second (orange) cycle for the AL and AC group. Letters indicate significant differences after pair t-test comparisons among groups within and between cycles for a *p*-value of 0.05.

### 3.3. ISSR

The number of amplified loci using six different ISSR primers was 71 during the first cycle (being 54 of them polymorphic, i.e 76,06 %) for all plants and 94 for the second cycle (being 75 of them polymorphic, i.e 79,79%) for all plants. The bands were considered to be polymorphic when its present frequency was between 5% and 95% among the individuals. Among the polymorphic bands, during the first cycle, 53 were shown by some of the AL plants and 54 by some of the AC plants. During the second cycle, 72 were showed by some of the AL plants and 75 by some AC plants. Figure 4 shows the constructed UPGMA dendrograms based on ISSR band amplifications before and after the pH substratum change. The results indicated that band profile changed with the treatment since sample clustering was different during the first and second cycle. In Figure 5A, plants destined for AL treatment and those for the AC treatment were randomly distributed in the dendrogram, while they separately clustered, in accordance with pH change after the second growth cycle.

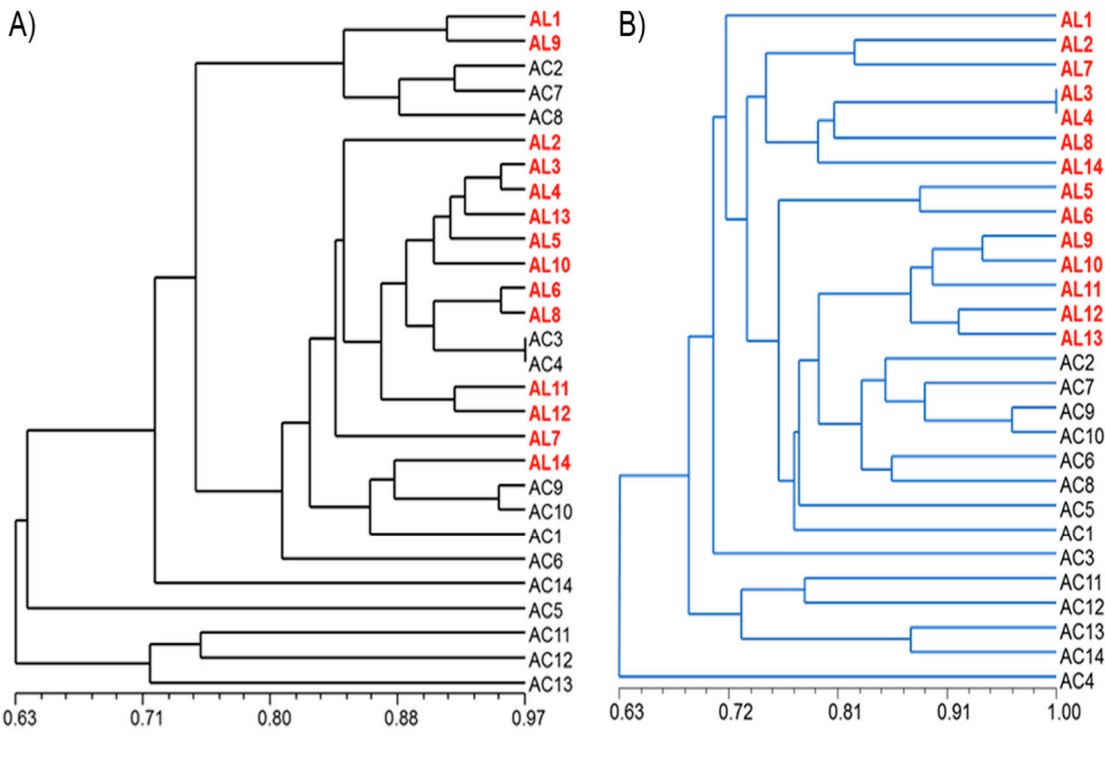

**Figure 5.** Unweighted Pair Group with Arithmetic Average (UPGMA) dendrograms based on the similarity Jaccard matrix constructed from the binary ISSR band matrices for the (**A**) first and (**B**) second cycle.

### 3.4. Mantel Test

The comparison among matrixes of Jaccard coefficient from ISSR and MSAP performed with the Mantel test reveals the correlation levels among treatments that are shown in Table 3. An increased correlation was found between data from ISSR in the first cycle and hemimethylated sites ($r = 0.359$) to the second cycle ($r = 0.460$) as well as the internal cytosine methylation that varies from 0.367 in first cycle to 0.408 in second cycle.

**Table 3.** Values for coefficient of correlation (*r*) produced with Mantel test.

| | ISSR 1st Cycle | ISSR 2nd Cycle | HpaII+/MspI− 1st Cycle | HpaII−/MspI+ 1st Cycle | HpaII+/MspI− 2nd Cycle | HpaII−/MspI+ 2nd Cycle |
|---|---|---|---|---|---|---|
| ISSR 1st cycle | 0 | | | | | |
| ISSR 2nd cycle | 0.283 | 0 | | | | |
| HpaII+/MspI− 1st cycle | 0.359 | 0.367 | 0 | | | |
| HpaII−/MspI+ 1st cycle | 0.042 | 0.133 | 0.089 | 0 | | |
| HpaII+/MspI− 2nd cycle | 0.460 | 0.408 | 0.689 | 0.123 | 0 | |
| HpaII−/MspI+ 2nd cycle | 0.233 | −0.0221 | 0.609 | 0.029 | 0.121 | 0 |

## 4. Discussion

The objective of the present work was to corroborate the change of the hydrangea colour sepals when the pH of the substratum is modified and relate these changes with the DNA methylation as an epigenetic mechanism. As already reported, the more acidic the substratum is, the bluer the inflorescence becomes as a result of different Al concentration in the vacuole in combination with the anthociyanine delphinidin 3-glucoside and different co-pigments [12,16–20]. Here, although the plants were already in an acid media before the pH change (Figures 1 and 2), the colour of the sepals changed for each plant after the substratum modification; however, groups could not be differentiated based on the colour after the second growth cycle (Figure 3). We did not found blue and red blooms due to the acid or alkaline treatment, as expected. Perhaps if we had studied more growth cycles, we could have found these expected colour differences. Nevertheless, after both treatments, the number of methylated DNA sites, as evidenced by the MSAP technique increased. Although this averaged number tends to be higher with the acid treatment, significant differences were not found among both of the groups with a t-test analysis due to the high dispersion presented. DNA methylation is an epigenetic mechanism that living organisms use for adaptations. It plays a role in diverse plant physiological processes, such as transcriptional regulation, vernalization, or long-term adaptation to the environment [31]. The exposure to acidy or alkaline substratum very likely determines a cascade of reactions in the hydrangea plants to face these new growth conditions. As said above, Chen et al. [24] reported several genes that were up or down regulated after analyzing the transcriptome of root or leaf cells after Al exposure and some Al transporters have already been described [22,23]. It is likely that these genetic regulations could, at least partially, be mediated by DNA methylation. Colour change has been related with DNA methylation of different genes and promoters in other species [25–28]. The MSAP used in present work has been useful in detecting methylated sites that are related to the treatments done, even if the CCGG recognition site can theoretically occur in 10 differentially forms [43]. Unmethylated CCGG sequences (shown by a HpaII+ /MspI+ pattern) were the most frequent ones in the first cycle, and it decreased in the second cycle, mainly in the AC group. The hemi or fully methylated CmCGG secuenques (shown by a HpaII-/MspI+ pattern) became more frequent during the second cycle, especially, again, in the AC group. According to Pérez-Figueroa et al. [41], an HpaII+ /MspI- pattern would mean a hemimethylated mCCGG or mCmCGG sequences, which followed the same increase as before. To end, and again according to Pérez-Figueroa et al. [41], the absence of both bands would occur when both of the cytosines are fully methylated or when a mutation has occurred. Interestingly, this type of methylation seemed to slightly decrease during the second cycle. Nevertheless, FulneČek, J. and Kovařík [43] reported ten different methylated forms, with some of them not being distinguished by the classical MspI/HpaII activity assays and suggest adding a combine digestion of both enzymes to help interpret those ambiguities. These should be taken into account in future works. The Mantel test demonstrates a correlation among the MSL found in first and second cycles and the values were increased between cycles, probably due to the methylated sites. The ISSR technique was also capable of detecting variations ascribable to alteration of the DNA methylation status. The ISSR results were surprisingly interesting. The aim of using this type of molecular marker was to assess the genetic stability of hydrangea plants after a pH substratum change. This type of marker has been previously used to check the genetic fidelity among mother and micropropagated plants in, for example, bamboo [44], *Stevia rebaudiana* [37], or *Ochreinauclea missionis* [36]. ISSR that

amplified by PCR inter-SSR sequences, are able to detect polymorphisms among different samples both when the priming sites are present/absent or when the length of the amplified sequence varied [45]. Here, it was observed not only a different DNA band profile before and after pH change of the growth substrate, but also differences for AC and AL plants. The UPGMA trees that were constructed based on the Jaccard similarity matrices were very clear: while during the first cycle, AL and AC plants were admixed, clustering separately during the second cycle (Figure 5). On one hand, differences among samples could be due to different amplicons lengths. This could be explained by transposons movements. Nishijima et al. [46] reported the existence of transposons that are inserted between R2R3-MYB transcription factors, which participated in the promotion of anthocyanin biosynthesis in torenia petals (*Torenia fournieri*). The insertion produces a mutant that fades the colour of the petals from violet to pale violet. We could not explain the ISSR property to differentiate between the two groups after the second cycle, since the transposon movement is, a priori, random. On the other hand, differences among ISSR bands profiles found could be due to the ability (or inability) of the primer to hybridize with its SSR complement sequence when cytosines are demethylated (or methylated).

In the present study we have demonstrated DNA methylation alterations in *Hydrangea macrophyla* when the pH of the substrate is modified. However, it is necessary to evaluate the expression of genes related to anthocyanin biosynthesis or Al assimilation in order to demonstrate whether methylation affects the colour of sepals. The data presented above will contribute to a better understanding of the molecular mechanisms that underlie the process of pigmentation changes in *Hydrangea macrophylla*.

## 5. Conclusions

MSAP is a powerful technique that can be used correlated with ISSR to detect epigenetic changes produced by environmental factors in plants like *H. macrophylla*. In addition, the ISSR can detect differences in treatments that come from different environments. However, it is necessary experimentation including more factors and genome sequencing in order to determinate diversity of cytosine methylation correlated with individual phenotypic traits.

**Author Contributions:** M.I.T.-M. and J.Y.A.-C. designed and conducted the experiments. N.A.-A., R.R.-M. and M.E.-D. participated in methodological proposal and collecting data. M.I.T.-M., J.Y.A.-C. and N.L. analyzed the data and wrote the paper.

**Funding:** J.Y.A.-C. was funded by CONACYT in BIMARENA program. The Mexican Government through the Mexican Agency funds N.L. for International Development Cooperation.

**Acknowledgments:** Authors acknowledge the administrative support by Departamento de Producción Agrícola CUCBA-UDG.

**Conflicts of Interest:** The authors declare no conflict of interest.

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
