# Peer review of "Hydrangea DNA Methylation Caused by pH Substrate Changes to Modify Sepal Colour is Detected by MSAP and ISSR Markers"

_agronomy, doi:10.3390/agronomy9120871_

Round 1
Reviewer 1 Report
This new version of the manuscript entitled “Hydrangea DNA methylation caused by pH substrate changes to modify sepal colour is detected by MSAP and ISSR markers” has improved its quality following almost all the previous comments. Nevertheless, the discussion of the research performed has to be improved as I stated in my previous comment to make the discussion more rigorous.
Specifically, in line 304-305 some states are not correct (“DNA methylation is an epigenetic mechanism that living organisms use for environmental adaptations. It has been observed when plants are exposed to biotic or abiotic stress [31]”). DNA methylation is a very complex mechanism with large physiological implications, being one of them the environmental adaptation but not the only one.
If authors want to talk about methylation and adaptation, they must add to the manuscript a discussion about methylation contexts. As I recommend to the authors in my previous revision, CG and CHG contexts (since CHH context are not studied in this assay) must be (at least) mentioned during their interpretation of the results. I encourage the authors to carefully read and use the information contained in the two references I kindly provide in my previous review. The relation between their MSAP assay and the different contexts (CG and CHG) could be found in the first one, and the implication of the different contexts and environmental adaptation in the second one: (Fulneček J, Kovařík A. How to interpret methylation-sensitive amplified polymorphism (MSAP) profiles?. BMC Genet. 2014;15:2. Published 2014 Jan 6. doi:10.1186/1471-2156-15-2; and: Sanchez‐Muñoz, R. , Moyano, E. , Khojasteh, A. , Bonfill, M. , Cusido, R. M. and Palazon, J. (2019), Genomic methylation in plant cell cultures: a barrier to the development of commercial long‐term biofactories. Eng. Life Sci.. Accepted Author Manuscript. doi:10.1002/elsc.201900024)).
Author Response
Response to Reviewer 1 Comments
Point 1: Specifically, in line 304-305 some states are not correct (“DNA methylation is an epigenetic mechanism that living organisms use for environmental adaptations. It has been observed when plants are exposed to biotic or abiotic stress [31]”). DNA methylation is a very complex mechanism with large physiological implications, being one of them the environmental adaptation but not the only one.
Response 1: Sentence was changed for this other one sentence in manuscript: “DNA methylation is an epigenetic mechanism that living organisms use for adaptations. It plays a roll in diverse plant physiological processes such as transcriptional regulation, vernalization or long-term adaptation to the environment”
Point 2: If authors want to talk about methylation and adaptation, they must add to the manuscript a discussion about methylation contexts. As I recommend to the authors in my previous revision, CG and CHG contexts (since CHH context are not studied in this assay) must be (at least) mentioned during their interpretation of the results. I encourage the authors to carefully read and use the information contained in the two references I kindly provide in my previous review. The relation between their MSAP assay and the different contexts (CG and CHG) could be found in the first one, and the implication of the different contexts and environmental adaptation in the second one:
Response 2: We analyse the references and trying to understand the interpretation of the concepts the reviewer kindly give us, we add this paragraph to the discussion.
“The MSAP used in present work has been useful to detect methylated sites related to the treatments done. Even if the CCGG recognition site can theoretically occur in 10 differentially forms [43]. Unmethylated CCGG sequences (shown by a HpaII + /MspI+ pattern) were the most frequent ones in the first cicle, and it decresed in the second cicle, mainly in the AC group. The hemi or fully methylated CmCGG secuenques (shown by a HpaII-/MspI+ pattern) became more frequent during the second cycle, specially, again, in the AC group. Acording to Pérez-Figueroa et al [41], an HpaII + /MspI- pattern would mean a hemimethylated mCCGG or mCmCGG sequences, that followed the same increase as before. To end, and again according to Pérez-Figueroa et al [41], the absence of both bands would occur when both cytosines are fully methylated or when a mutation has occured. Interestingly, this tipe of methylation seemed to slightly decreaseduring the second cicle. Nevertheless, FulneČek, J. and Kovařík [43] reported ten different methylated forms, some of them being not distinguish by the clasical MspI/HpaII activity assays and suggest adding a combine digestion of both enzimes to help to interpret those ambiguities. These should be taken into account in future works”
Point 3: (Fulneček J, Kovařík A. How to interpret methylation-sensitive amplified polymorphism (MSAP) profiles?. BMC Genet. 2014;15:2. Published 2014 Jan 6. doi:10.1186/1471-2156-15-2; and: Sanchez‐Muñoz, R. , Moyano, E. , Khojasteh, A. , Bonfill, M. , Cusido, R. M. and Palazon, J. (2019), Genomic methylation in plant cell cultures: a barrier to the development of commercial long‐term biofactories. Eng. Life Sci.. Accepted Author Manuscript. doi:10.1002/elsc.201900024)).
Response 3: These references are now included in the manuscript
Reviewer 2 Report
Please see minor modifications I have done of the text

Author Response
Response to Reviewer 2 Comments
We the authors want to thank you for your suggestions
We want to thank you for your suggestions and we have made the changes you indicated in the text of the article.
This manuscript is a resubmission of an earlier submission. The following is a list of the peer review reports and author responses from that submission.
Round 1
Reviewer 1 Report
The paper entitled “Hydrangea DNA methylation caused by pH substrate changes to modify sepal colour is detected by MSAP and ISSR markers” studies the relation between the phenotype of different hyndrangea cultivars with the methylation patterns in front of pH changes. To do that, MSAP and ISSR methodology was used. I believe that this preliminary study provides the clues to a better understanding of molecular regulation of pigmentation in Hydrangea macrophylla. Despite this, in order to consider the present manuscript to be published, some implementations must be done.
Major changes:
The main change must be done in the interpretation of the results. In the graphs, the mean and standard deviation were used to represent a group of N=14 in terms of pH and methylation. There are not significant differences showed in terms of methylation differences between groups. As the authors are referring to methylation increase, the best way to represent this data is giving the methylation increase between the same plant under different conditions (P1 during the first cycle compared with P1 during the second cycle). Then, the mean and SD could be done in order to find differences. It could be one of the causes to not find significant differences between groups (as methylation are very dynamic events and the methylation levels between plants could not be exactly the same). On the other hand, the use of the enzyme MspI and HpaII allow to study only CG and CHG methylated cytosines, and they do not cover the CHH ones (that are shown to be the most dynamic context). In fact, the comparison between the patterns shown by the MspI enzyme and the HpaII one, allow to differentiate between CG and CHG methylation (a certain approximation). Although I know that it was not the purpose of this study, I think that some kind of discussion about this topic must be added to the discussion/conclusion of the manuscript (I strongly recommend to explore the references: Fulneček J, Kovařík A. How to interpret methylation sensitive amplified polymorphism (MSAP) profiles?. BMC Genet. 2014;15:2. Published 2014 Jan 6. doi:10.1186/1471-2156-15-2; and: Sanchez‐Muñoz, R. , Moyano, E. , Khojasteh, A. , Bonfill, M. , Cusido, R. M. and Palazon, J. (2019), Genomic methylation in plant cell cultures: a barrier to the development of commercial long‐term biofactories. Eng. Life Sci.. Accepted Author Manuscript. doi:10.1002/elsc.201900024). In terms of pH and pigmentation changes, I recommend representing the data pointing out the pH difference in the plot were the color changes are indicated. With all this data, the results section could be improved.
Minor changes:
In all the plots, the legend of the axis must be added. In the bar plots, the positive standard deviation must be added. Some typographic mistakes must be corrected: This species ability to (Line 42); The biosystesis (Line 48); mediated by biosystethetic (Line 49), etc.
Reviewer 2 Report
Please consider comments, advices and suggestions introduced in the body of the text (see the attached file).
